# Triapine Derivatives Act as Copper Delivery Vehicles to Induce Deadly Metal Overload in Cancer Cells

**DOI:** 10.3390/biom10091336

**Published:** 2020-09-19

**Authors:** Kateryna Ohui, Iryna Stepanenko, Iuliana Besleaga, Maria V. Babak, Radu Stafi, Denisa Darvasiova, Gerald Giester, Vivien Pósa, Eva A. Enyedy, Daniel Vegh, Peter Rapta, Wee Han Ang, Ana Popović-Bijelić, Vladimir B. Arion

**Affiliations:** 1Institute of Inorganic Chemistry, University of Vienna, Währinger Strasse 42, A-1090 Vienna, Austria; kateryna.ohui@univie.ac.at (K.O.); besleaga.iuliana04@gmail.com (I.B.); radustafi@gmail.com (R.S.); 2Department of Chemistry, National University of Singapore, 3 Science Drive 2, Singapore 117543, Singapore; ang.weehan@nus.edu.sg; 3Department of Chemistry, City University of Hong Kong, 83 Tat Chee Avenue, Hong Kong SAR 999077, China; 4Institute of Physical Chemistry and Chemical Physics, Slovak University of Technology in Bratislava, Radlinského 9, SK-81237 Bratislava, Slovakia; denisa.darvasiova@stuba.sk (D.D.); peter.rapta@stuba.sk (P.R.); 5Department of Mineralogy and Crystallography, University of Vienna, Althan Strasse 14, A-1090 Vienna, Austria; gerald.giester@univie.ac.at; 6Department of Inorganic and Analytical Chemistry, Interdisciplinary Excellence Centre, University of Szeged, Dóm tér 7, H-6720 Szeged, Hungary; posavivien@gmail.com (V.P.); enyedy@chem.u-szeged.hu (E.A.E.); 7MTA-SZTE Lendület Functional Metal Complexes Research Group, University of Szeged, Dóm tér 7, H-6720 Szeged, Hungary; 8Institute of Organic Chemistry, Catalysis and Petrochemistry, Department of Organic Chemistry, Slovak University of Technology in Bratislava, Radlinského 9, SK-81237 Bratislava, Slovakia; daniel.vegh@stuba.sk; 9Faculty of Physical Chemistry, University of Belgrade, 11158 Belgrade, Serbia; ana@ffh.bg.ac.rs

**Keywords:** triapine, amidrazones, isothiosemicarbazones, copper(II), iron(III), cancer signalling

## Abstract

Thiosemicarbazones continue to attract the interest of researchers as potential anticancer drugs. For example, 3-aminopyridine-2-carboxaldehyde thiosemicarbazone, or triapine, is the most well-known representative of this class of compounds that has entered multiple phase I and II clinical trials. Two new triapine derivatives **HL^1^** and **HL^2^** were prepared by condensation reactions of 2-pyridinamidrazone and S-methylisothiosemicarbazidium chloride with 3-*N*-(*tert*-butyloxycarbonyl) amino-pyridine-2-carboxaldehyde, followed by a Boc-deprotection procedure. Subsequent reaction of **HL^1^** and **HL^2^** with CuCl_2_·2H_2_O in 1:1 molar ratio in methanol produced the complexes **[Cu^II^(HL^1^)Cl_2_]·H_2_O** (**1·H_2_O**) and **[Cu^II^(HL^2^)Cl_2_]** (**2**). The reaction of **HL^2^** with Fe(NO_3_)_3_∙9H_2_O in 2:1 molar ratio in the presence of triethylamine afforded the complex **[Fe^III^(L^2^)_2_]NO_3_∙0.75H_2_O** (**3∙0.75H_2_O**), in which the isothiosemicarbazone acts as a tridentate monoanionic ligand. The crystal structures of **HL^1^**, **HL^2^** and metal complexes **1** and **2** were determined by single crystal X-ray diffraction. The UV-Vis and EPR spectroelectrochemical measurements revealed that complexes **1** and **2** underwent irreversible reduction of Cu(II) with subsequent ligand release, while **3** showed an almost reversible electrochemical reduction in dimethyl sulfoxide (DMSO). Aqueous solution behaviour of **HL^1^** and **1,** as well as of **HL^2^** and its complex **2**, was monitored as well. Complexes **1**−**3** were tested against ovarian carcinoma cells, as well as noncancerous embryonic kidney cells, in comparison to respective free ligands, triapine and cisplatin. While the free ligands **HL^1^** and **HL^2^** were devoid of antiproliferative activity, their respective metal complexes showed remarkable antiproliferative activity in a micromolar concentration range. The activity was not related to the inhibition of ribonucleotide reductase (RNR) R2 protein, but rather to cancer cell homeostasis disturbance—leading to the disruption of cancer cell signalling.

## 1. Introduction

α-*N*-Heterocyclic thiosemicarbazones (TSCs) attract considerable attention since their remarkable antiproliferative properties have been discovered [1]. The most promising representative of this group of compounds is 3-aminopyridine-2-carboxaldehyde thiosemicarbazone or triapine (3-AP), which has already entered more than 30 phase I and II clinical trials [2,3,4]. 3-AP was also tested in various combination regimens and was shown to enhance the activity of other anticancer drugs, such as cisplatin [2,5], gemcitabine [6], doxorubicin [7], irinotecan [4]. It is believed that the mode of action of 3-AP is based on the effective inhibition of RNR [8,9,10] as a result of Fe(II) chelation from the diferric-tyrosyl cofactor, followed by the generation of reactive oxygen species (ROS) [11]. However, despite the promising anticancer effects in vitro and in vivo, the use of 3-AP in patients is associated with side-effects, i.e., neutropenia [12], methemoglobinemia [13], nausea, vomiting etc. Additionally, several types of solid tumours demonstrated resistance to 3-AP treatment [12,14,15,16]. TSCs are known to efficiently chelate first-row transition metals, 3-AP, and its analogues were coordinated to endogenous metals, such as Cu and Fe [17,18,19,20]. TSC coordination to essential metals often decreases overall toxicity of the drug molecules [21] and increases their anticancer activity, due to the synergistic action of TSCs and the metal centre [19,20,21]. In general, numerous Cu(II) complexes of thiosemicarbazones were characterised by superior anticancer activity when compared to that of the free ligands [22,23,24,25,26]. In contrast, coordination to Fe(III) prior to the administration of the compound did not always result in the improvement of cytotoxicity [17,27]. Interestingly, both Cu(II) and Fe(III) complexes of 3-AP were less active than 3-AP itself [10,17], possibly due to their high thermodynamic stability and lower cell uptake kinetics [28].

Carboxamidrazones structurally resemble thiosemicarbazones, but, surprisingly, their anticancer effects were not extensively studied. These compounds demonstrated antiproliferative activity in the micromolar range and good selectivity to cancer cells over healthy cells [29,30]. Similarly to Cu(II)-TSC complexes, coordination of carboxamidrazones to Cu(II) improved the cytotoxicity. However, there are only several examples of reported Cu(II)-carboxamidrazone complexes [31,32,33]. Therefore, we have prepared novel pyridinamidrazone and S-methylisothiosemicarbazone, structurally related to 3-AP, as well as Cu(II) and Fe(III) complexes thereof (Scheme 1) and investigated their potential as anticancer drugs. It should be noted that thiomethylation is known to affect the coordination behaviour of isothiosemicarbazones toward first-row transition metals when compared to thiosemicarbazones, so that sulfur atom is not involved in binding to the metal [34]. It might also have an effect not only on the antiproliferative activity, but also on the underlying mechanism of action; therefore, one of the main aims was to address this issue in the present study. All compounds were fully characterised by analytical and spectroscopic methods. The electrochemical properties were studied by cyclic voltammetry and UV–Vis–NIR spectroelectrochemistry. Solution stabilities of **HL^1^** and **1**, as well as of **HL^2^** and **2** were monitored by UV–Vis spectroscopy. The anticancer activity of novel compounds was tested in two ovarian cancer cell lines (A2780 and A2780cis), as well as in noncancerous embryonic kidney cell line (HEK293) and their intracellular dose-dependent accumulation investigated. The ability of **HL^2^** to quench the tyrosyl radical in hR2 RNR protein and of its Fe(II) complex to generate ROS in aqueous solution, was studied as well and compared with that of 3-AP and its Fe(II) complex. In addition, the effects of the most cytotoxic complex **2** on the key cancer cell signalling pathways were described.

## 2. Materials and Methods

### 2.1. Chemicals

All reagents used were received from commercial sources. 2-Cyanopyridine, CuCl_2_·2H_2_O and Fe(NO_3_)_3_·9H_2_O were purchased from Sigma-Aldrich. 2-Pyridinamidrazone was prepared as a white solid by the reaction of 2-cyanopyridine with excess hydrazine monohydrate as reported previously [35]. 3-*N*-(*tert*-Butyloxycarbonyl)amino-2-pyridinecarboxaldehyde [36] and S-methylisothiosemicarbazidium iodide [37] were obtained, as described in the literature. The S-methylisothiosemicarbazidium chloride was obtained from the corresponding iodide by using anion-exchange resin.

#### 2.1.1. 3-Amino-2-pyridinecarboxaldehyde 2-pyridinamidrazone (**HL^1^**)

To a solution of 3-*N*-(*tert*-butyloxycarbonyl)amino-2-pyridinecarboxaldehyde (0.21 g, 0.945 mmol) and 2-pyridinamidrazone (0.13 g, 0.945 mmol) in ethanol (8 mL) 12 M HCl (0.079 mL, 0.948 mmol) was added dropwise. The resulting solution was stirred at room temperature for 2 h, and then evaporated under reduced pressure to produce an orange oil. This was crystallised in vacuo to give **Boc-HL^1^**·HCl (C_17_H_20_N_6_O_2_·HCl, positive ion ESI-MS (MeCN/MeOH+1% H_2_O): *m/z* 341.16 [Boc-HL^1^+H]^+^). A part of the raw product (0.25 mg, 0.66 mmol) was suspended in ethyl acetate (3 mL) and a solution of 4 M HCl/EtOAc (1 mL, 4 mmol) was added. The resulting yellow suspension was stirred at room temperature for 1 h and then at 45 °C for 2 h. The Boc-deprotection was monitored by ESI-MS (positive ion ESI-MS (MeCN/MeOH+1% H_2_O): *m/z* 241.23 [HL^1^+H]^+^). The suspension was neutralised with saturated NaHCO_3_ (pH 7–8), then the product was extracted with ethyl acetate, washed with brine, dried over Na_2_SO_4_ and evaporated in vacuo to produce a yellow powder of **HL^1^** (C_12_H_12_N_6_, yield 0.12 g, 0.49 mmol, 74%). Crystals suitable for X-ray diffraction study were obtained in ethyl acetate/hexane mixture (1/1 *v*/*v*). Anal. Calcd for C_12_H_12_N_6_ (*M_r_* = 240.26), %: C, 59.98; H, 5.03; N, 34.98. Found, %: C, 59.78; H, 5.15; N, 34.65. Positive ion ESI-MS for C_12_H_12_N_6_ (MeCN/MeOH+1% H_2_O): *m/z* 241.23 [HL^1^+H]^+^, 263.22 [HL^1^+Na]^+^, negative ion ESI-MS: *m/z* 239.09 [HL^1^–H]^–^. ^1^H NMR (600 MHz, DMSO-d_6_) δ, ppm: 8.66 (d, *J* = 4.7 Hz, 1H, H_6_), 8.63 (s, 1H, H_10_), 8.28 (d, *J* = 7.9 Hz, 1H, H_3_), 7.92 (td, *J* = 7.8, 1.7 Hz, 1H, H_4_), 7.88 (dd, *J* = 4.2, 1.3 Hz, 1H, H_13_), 7.54 (ddd, *J* = 7.4, 4.8, 1.0 Hz, 1H, H_5_), 7.20 (dd, *J* = 8.3, 1.0 Hz, 1H, H_15_), 7.10 (brs, 2H, H_7’_), 7.09 (dd, *J* = 8.3, 4.3 Hz, 1H, H_14_), 6.85 (s, 2H, H_16’_). ^13^C NMR (151 MHz, DMSO-d_6_) δ, ppm: 160.17 (C_10_), 155.03 (C_7_), 150.42 (C_2_), 148.42 (C_6_), 144.93 (C_16_), 137.09 (C_4_ or C_13_), 137.01 (C_4_ or C_13_), 133.90 (C_11_), 125.36 (C_5_), 124.31 (C_14_), 121.94 (C_15_), 121.28 (C_3_). ^15^N NMR (61 MHz, DMSO-d_6_) δ, ppm: 349.66 (N_8_), 321.97 (N_12_), 304.04 (N_1_), 275.94 (N_9_), 76.02 (N_7′_), 71.27 (N_16′_). The atom numbering for the assignment of resonances see in Appendix A.

#### 2.1.2. **[Cu(HL^1^)Cl_2_]·H_2_O**∙(**1·H_2_O**)

To a solution of **HL^1^** (0.100 g, 0.42 mmol) in methanol (10 mL) was added a solution of copper(II) chloride dihydrate (0.062 g, 0.36 mmol) in methanol (10 mL). The mixture was heated to 65 °C and left under stirring and reflux overnight. Next day the brown precipitate was filtered off and purified on preparative HPLC by using water/methanol as eluent (Appendix A). The final product was obtained as brown powder after drying in vacuo. Yield: 0.031 g, 22%. Anal. Calcd for C_12_H_12_Cl_2_CuN_6_·H_2_O (*M_r_* = 392.73), %: C, 36.69; H, 3.59; N, 21.39. Found, %: C, 37.00; H, 3.32; N, 21.13. Positive ion ESI-MS for C_12_H_12_Cl_2_CuN_6_ (MeCN/MeOH+1% H_2_O): *m/z* 338.14 [Cu(HL^1^)Cl]^+^, 303.17 [Cu^I^(HL^1^)]^+^ (Appendix A), negative ion ESI-MS: *m*/*z* 336.02 {[Cu(HL^1^)Cl]^+^–2H^+^}^–^, 373.98 [Cu(HL^1^)Cl_2_–H^+^]^–^. UV˗Vis (DMSO), λ_max_, nm (ε, M^−1^cm^−1^): 714 (139), 437 (10,326). IR (ATR, selected bands, ῦ_max_): 3209, 3114, 1636, 1578, 1538, 1235, 1151, 1052, 853, 801, 683, 660 cm^−1^.

#### 2.1.3. 3-Amino-2-pyridinecarboxaldehyde S-methylisothiosemicarbazone (**HL^2^**)

3-*N*-(*tert*-Butyloxycarbonyl)amino-2-pyridinecarboxaldehyde (0.31 g, 1.4 mmol) in ethanol (6 mL) and S-methylisothiosemicarbazidium chloride (0.20 g, 1.4 mmol) in water (2 mL) were mixed together and 12 M HCl (0.597 mL) was added. The mixture was stirred at 70 °C for 3 h. After cooling to the room temperature the solvent was removed under reduced pressure, the residue dissolved in water and neutralised with NaHCO_3_. Fine yellow precipitate was filtered off, washed with ethanol, diethyl ether and dried in vacuo. Yield: 0.19 g, 67%. Anal. Calcd for C_8_H_11_N_5_S (*M_r_* = 209.27): C, 45.91; H, 5.30; N, 33.47; S, 15.32%. Found, %: C, 46.07; H, 5.28; N, 33.29; S, 15.46. Positive ion ESI-MS for C_8_H_11_N_5_S (MeCN/MeOH+1% H_2_O): *m/z* 210.06 [HL^2^+H]^+^. ^1^H NMR (600 MHz, DMSO-d_6_, *Z*-isomer) δ, ppm: 8.39 (s, 1H, H_8_), 7.83 (dd, *J* = 4.3, 1.5 Hz, 1H, H_6_), 7.13 (dd, *J* = 8.3, 1.1 Hz, 1H, H_4_), 7.04–7.01 (m, 2H, H_5_ (*Z*-)+H_5_ (*E*-)), 6.90 (s, 2H, H_13_), 6.67 (s, 2H, H_7_), 2.39 (s, 3H, H_12_). ^1^H NMR (600 MHz, DMSO-d_6_, *E*-isomer) δ, ppm: 8.30 (s, 1H, H_8_), 7.82 (dd, *J* = 4.3, 1.5 Hz, 1H, H_6_), 7.08 (dd, *J* = 8.3, 1.1 Hz, 1H, H_4_), 7.04–7.01 (m, 2H, H_5_ (*Z*-)+H_5_ (*E*-)), 6.87 (s, 2H, H_7_), 6.82 (s, 2H, H_13_), 2.40 (s, 3H, H_12_). ^13^C NMR (151 MHz, DMSO-d_6_, *Z*-isomer) δ, ppm: 160.43 (C_11_), 157.86 (C_8_), 144.45 (C_3_), 136.88 (C_6_), 134.19 (C_2_), 123.85 (C_5_), 121.65 (C_4_), 12.30 (C_12_). ^13^C NMR (151 MHz, DMSO-d_6_, *E-*isomer) δ, ppm: 165.50 (C_11_), 153.74 (C_8_), 144.00 (C_3_), 136.82 (C_6_), 134.70 (C_2_), 123.41 (C_5_), 121.52 (C_4_), 12.50 (C_12_). ^15^N NMR (61 MHz, DMSO-d_6_, *Z*-isomer) δ, ppm: 70.93 (N_7_), 83.70 (N_13_), 267.43 (N_10_), 321.37 (N_1_), 348.34 (N_9_). ^15^N NMR (61 MHz, DMSO-d_6_, *E*-isomer) δ, ppm: 66.65 (N_7_), 80.46 (N_13_), 280.47 (N_10_), 319.87 (N_1_), 357.71 (N_9_).

#### 2.1.4. **[Cu(HL^2^)Cl_2_]** (**2**)

To a solution of **HL^2^** (0.050 g, 0.24 mmol) in methanol (5 mL) was added a solution of copper(II) chloride dihydrate (0.041 g, 0.24 mmol) in methanol (5 mL). The mixture was heated to 50 °C, stirred for 10 min, and allowed to cool down to room temperature. Dark-green crystals were filtered off next day, washed with small portions of cold methanol and dried in air. Yield: 0.034 g, 41%. Anal. Calcd for C_8_H_11_Cl_2_CuN_5_S (*M_r_* = 343.72), %: C, 27.95; H, 3.23; N, 20.37; S, 9.33. Found, %: C, 28.13; H, 3.30; N, 19.89; S, 8.92. Positive ion ESI-MS for C_8_H_11_Cl_2_CuN_5_S (MeCN/MeOH+1% H_2_O): *m*/*z* 271.15 [Cu^II^(L^2^)]^+^; 307.13 [Cu(HL^2^)Cl]^+^; negative: *m*/*z* 304.95 [Cu(L^2^)Cl–H^+^]^–^. UV-Vis (MeOH), λ_max_, nm (ε, M^−1^cm^−1^): 696 (122), 441 (11643), 282 (12217). IR (ATR, selected bands, ῦ_max_): 3250, 1642, 1583, 1525, 1502, 1274, 1217, 913, 877, 794, 643 cm^−1^.

#### 2.1.5. **[Fe(L^2^)_2_NO_3_]∙0.75H_2_O** (**3∙0.75H_2_O**)

To a solution of **HL^2^** (0.059 g, 0.28 mmol) in methanol (5 mL) was added a solution of Fe(NO_3_)_3_·9H_2_O (0.057 g, 0.14 mmol) in methanol (5 mL) and triethylamine (39 μL, 0.28 mmol). The mixture was heated to 60 °C and stirred for 20 min. After the removal of methanol under reduced pressure, the residue was recrystallised from ethanol to give fine black microcrystals. These were filtered off, washed with ethanol, diethyl ether and dried in air. Yield: 0.033 g, 43%. Anal. Calcd for C_16_H_20_FeN_11_O_3_S_2_∙0.75H_2_O (*M_r_* = 547.9), %: C, 35.07; H, 3.96; N, 28.12; S, 11.71. Found, %: C, 35.36; H, 3.73; N, 27.90; S, 11.64. Positive ion ESI-MS for C_16_H_20_FeN_11_O_3_S_2_ (MeCN/MeOH+1% H_2_O): *m/z* 472.20 [Fe(L^2^)_2_]^+^ (Appendix A); negative ion: *m*/*z* 470.03 [Fe(L^2^)_2_–2H^+^]^–^. UV–Vis (MeOH), λ_max_, nm (ε, M^−1^cm^−1^): 932 (891), 694 (590), 488 (12551), 428 (13956), 401sh, 301(15888), 266 (14206). IR (ATR, selected bands, ῦ_max_): 3214, 1646, 1579, 1528, 1466, 1319, 1268, 1142, 1057, 970, 849, 736 cm^−1^.

### 2.2. Physical Measurements

Elemental analyses were carried out in a Carlo-Erba microanalyser at the Microanalytical Laboratory of the University of Vienna. Electrospray ionisation mass spectrometry (ESI MS) was carried out with amaZon speed ETD Bruker instrument (Bruker Daltonik GmbH, Bremen, Germany, *m/z* range 0–900, ion positive/negative mode, 180 °C, heating gas N_2_ (5 L/min), Capillary 4500 V, End Plate Offset 500 V). UV–Vis spectra were measured on Perkin Elmer UV/Vis spectrophotometer Lambda 35. IR spectra were reported on a Bucker Vertex 70 Fourier transform IR spectrometer (4000–400 cm^−1^) using the ATR technique. 1D (^1^H, ^13^C) and 2D (^1^H-^1^H COSY, ^1^H-^1^H TOCSY, ^1^H-^1^H NOESY, ^1^H-^13^C HSQC, ^1^H-^13^C HMBC, ^1^H-^15^N HSQC, ^1^H-^15^N HMBC) NMR spectra were measured on a Bruker AV NEO 500 or AV III 600 spectrometers (Bruker BioSpin GmbH, Rheinstetten, Germany) in DMSO-d_6_ at 25 °C at NMR spectroscopy Centre of the Faculty of Chemistry of the University of Vienna.

### 2.3. Crystallographic Structure Determination

X-ray diffraction measurements were performed on a Bruker X8 APEXII CCD (Karlsruhe, Germany), Bruker D8 Venture (Karlsruhe, Germany) and STOE (Darmstadt, Germany) diffractometers. Single crystals were positioned at 30, 30, 35, 40 and 40 mm from the detector, and 987, 360, 3319, 5298 and 527 frames were measured, each for 20, 3, 10, 30 and 50 s over 0.5, 0.5, 0.5, 1 and 2° scan width for **HL^1^_,_ [H_3_L^1a^]Cl_2_·2H_2_O**, **HL^2^**, **1**, and **2**, respectively. The data were processed using SAINT software [38]. Crystal data, data collection parameters, and structure refinement details are given in Table 1. The structures were solved by direct methods and refined by full-matrix least-squares techniques. Non-H atoms were refined with anisotropic displacement parameters. H atoms were inserted in calculated positions and refined with a riding model. The following computer programs and hardware were used: structure solution, *SHELXS-2014* and refinement, *SHELXL-2014* [39]; molecular diagrams, ORTEP [40]; computer, Intel CoreDuo. CCDC 1958520, 1958521, 1958523–1958525.

### 2.4. Electrochemistry and Spectroelectrochemistry

Cyclic voltammetric experiments with 0.5 mM solutions of **1**–**3** in 0.1 M *n*-Bu_4_NPF_6_ (puriss quality from Fluka (Schwerte, Germany); dried under reduced pressure at 70 °C for 24 h before use) supporting electrolyte in DMSO (SeccoSolv max. 0.025% H_2_O, Merck) were performed under argon atmosphere using a three-electrode arrangement with a platinum disk or a glassy carbon disk working electrode (from Ionode, Australia), platinum wire as a counter electrode, and silver wire as a pseudo reference electrode. All potentials in voltammetric studies were quoted vs ferricenium/ferrocene (Fc^+^/Fc) redox couple. A Heka PG310USB (Lambrecht, Germany) potentiostat with a PotMaster 2.73 software package served for the potential control in voltammetric studies. In situ ultraviolet-visible-near-infrared (UV–Vis–NIR) spectroscopic and spectroelectrochemical measurements were performed on a spectrometer Avantes (Model AvaSpec-2048 × 14-USB2) in 1 cm quartz cuvette or the spectroelectrochemical cell kit (AKSTCKIT3) with the Pt-microstructured honeycomb working electrode, purchased from Pine Research Instrumentation (Lyon, France). The cell was positioned in the CUV-UV Cuvette Holder (Ocean Optics, Ostfildern, Germany) connected to the diode-array UV–Vis–NIR spectrometer by optical fibres. UV–Vis–NIR spectra were processed using the AvaSoft 7.7 software package. Halogen and deuterium lamps were used as light sources (Avantes, Model AvaLight-DH-S-BAL, Apeldoorn, The Netherlands).

### 2.5. Cell Lines and Culture Conditions

Human ovarian carcinoma cells A2780 and A2780cis, and human embryonic kidney cells HEK293 were obtained from ATCC. A2780 and A2780cis cells were cultured in RPMI 1640 medium containing 10% fetal bovine serum (FBS). HEK293 cells were cultured in DMEM medium containing 10% FBS. All cells were grown in tissue culture 25 cm^2^ flasks (BD Biosciences, Singapore) at 37 °C in a humidified atmosphere of 95% air and 5% CO_2_. All drug stock solutions were prepared in DMSO, and the final concentration of DMSO in the medium did not exceed 1% (*v*/*v*) at which cell viability was not inhibited. The amount of actual Cu concentration in the stock solutions was determined by ICP-OES.

### 2.6. Inhibition of Cell Viability Assay

The cytotoxicity of the compounds was determined by colourimetric microculture assay (MTT assay). The cells were harvested from culture flasks by trypsinisation and seeded into Cellstar 96-well microculture plates (Greiner Bio-One, Practical Mediscience Pte Ltd, Singapore, Singapore) at the seeding density of 6 × 10^4^ cells per mL. After the cells were allowed to resume exponential growth for 24 h, they were exposed to drugs at different concentrations in media for 72 h. The drugs were diluted in complete medium at the desired concentration, and 100 μL of the drug solution was added to each well and serially diluted to other wells. After exposure for 72 h, drug solutions were replaced with 100 μL of MTT in media (5 mg mL^−1^) and incubated for an additional 45 min. Subsequently, the medium was aspirated, and the purple formazan crystals formed in viable cells were dissolved in 100 μL of DMSO per well. Optical densities were measured at 570 nm with a microplate reader. The number of viable cells was expressed in terms of treated/control (T/C) values by comparison to untreated control cells, and 50% inhibitory concentrations (IC_50_) were calculated from concentration-effect curves by interpolation. The evaluation was based on means from at least three independent experiments, each comprising six replicates per concentration level.

### 2.7. Intracellular Accumulation

Intracellular accumulation of **1** and **2** was determined in A2780 cells. Cells were seeded into Cellstar 6-well plates (Greiner Bio-one) at a density of 6 × 10^5^ cells/well (2 mL per well). After the cells were allowed to resume exponential growth for 24 h, they were exposed to **1** and **2** at various concentrations for 24 h at 37 °C. The cells were washed twice with 1 mL of PBS and lysed with 100 μL of RIPA lysis buffer (ultrapure water, 5 M NaCl, 1 M Tris-HCl pH 8.0, 2% sodium deoxycholate, 10% SDS, IGEPAL, protease and phosphatase inhibitor) for 5–10 min at 4 °C. The cell lysates were scraped from the wells and transferred to separate 1.5 mL microtubes. The supernatant was then collected after centrifugation (13,000 rpm, 4 °C for 15 min) and the total protein content of each sample was quantified via a Bradford’s assay. Cell lysates were transferred to 2 mL glass vials and then digested with ultrapure 60% HNO_3_ (50 μL) at 110 °C for 96 h. The resulting solution was diluted to 1 mL (2–4% (*v**/v*) HNO_3_) with ultrapure Milli-Q water, sonicated for 45 min and filtered through 0.45 μm filters. Cu content of each sample was quantified by ICP-MS and expressed as per mg of protein. Re was used as an internal standard. Cu and Re were measured at *m*/*z* 64 and *m*/*z* 186, respectively. Metal standards for calibration curve (0, 1, 2, 5, 10, 20, 40 ppb) were prepared. All readings were performed in six replicates in He mode.

### 2.8. Spectrophotometric and pH-Potentiometric Measurements

Agilent Carry 8454 diode array spectrophotometer was used to record the UV–Vis spectra in the interval 200–800 nm. The path length was 1 or 2 cm. Solution stability of ligands **HL^1^** and **HL^2^** and their copper(II) complexes was monitored at various pH values by UV–Vis spectroscopy. The conditional stability constants (*K*’) of **1** and **2** were attempted to be calculated at pH 5.90 based on the spectral changes via the displacement reaction with ethylenediaminetetraacetic acid (EDTA) in the presence 50 mM 2-(N-morpholino)ethanesulfonic acid (MES) and 0.1 M KCl. In this competition experiment, the samples contained 50 μM complex and the concentration of EDTA was varied in the range from 0 to 422 μM. The reaction of **1** and **2** with glutathione (GSH) was studied at 25.0 ± 0.1 °C on Hewlett Packard 8452A diode array spectrophotometer using a special, tightly closed tandem cuvette (Hellma Tandem Cell, 238-QS, Altmann Analytik GmbH & Co. KG, Munich, Germany). The reactants were held separate until the reaction was triggered. Both isolated pockets of the cuvette were completely deoxygenated by bubbling a stream of argon for 10 min before mixing the reactants. Spectra were recorded before and then immediately after the mixing, and changes were followed until no further absorbance change was observed. One of the isolated pockets contained the reducing agent (12 mM GSH), and the other contained the studied compound (100 μM). The pH of all the solutions was adjusted to 7.40 by 50 mM (0.1 M KCl) HEPES buffer. The pH-potentiometric titrations were performed on samples containing **HL^2^** at 3 mM concentration with a KOH solution in the presence of 0.1 M KCl at 25.0 ± 0.1 °C. An Orion 710A pH-meter equipped with a Metrohm combined electrode (type 6.0234.100, Metrohm Inula GmbH, Vienna, Austria) and a Metrohm 665 Dosimat burette (Metrohm Inula GmbH, Vienna, Austria) were used for the pH-metric titrations. The electrode system was calibrated to the pH = −log[H^+^] scale using titrations of HCl with KOH according to the method suggested by Irving et al. [41]. The average water ionisation constant (p*K*_w_) is 13.76 ± 0.05 in water. Argon was also passed over the solutions during the titrations. UV–Vis titration was performed for **HL^1^**, which is less water-soluble compared to **HL^2^**, and the data were evaluated up to pH 6. Calculation of the p*K*_a_ values was performed with the computer programs PSEQUAD (**HL^1^**) and HYPERQUAD (**HL^2^**) [42].

### 2.9. Tyrosyl Radical Reduction in Human R2 RNR Protein

The time-dependent kinetics of tyrosyl radical destruction in human R2 ribonucleotide reductase protein (hR2) by 3-AP and **HL^2^** were measured by EPR spectroscopy at 30 K, on a Bruker Elexsys II E540 EPR spectrometer with an Oxford Instruments ER 4112HV helium cryostat. The experimental conditions were: microwave power 3.2 mW, modulation amplitude 5 G, modulation frequency 100 kHz, and conversion time 0.0293 s. Spectra were recorded and analysed using the Bruker Xepr software. The concentration of the tyrosyl radical was determined by double integration of EPR spectra recorded at non-saturating microwave power levels and compared with the copper standard [43]. Purified, recombinant, iron-reconstituted hR2 [44] was obtained from the Department of Biochemistry and Biophysics, Stockholm University, Sweden. The samples containing 20 μM hR2 in 50 mM Hepes buffer, pH 7.60/100 mM KCl/5% glycerol, and 20 μM 3-AP or **HL^2^** in 1% (*v*/*v*) DMSO/H_2_O, and 2 mM dithiothreiotol were incubated for indicated times and quickly frozen in cold isopentane. The same sample was used for repeated incubations at room temperature. The experiments were performed in duplicates.

### 2.10. EPR Spin Trapping Experiments

The generation of paramagnetic intermediates was monitored by cw-EPR spectroscopy using the EMX spectrometer (Bruker). The EPR spectra were measured with following experimental parameters: X-band, room temperature, microwave frequency, 9.431 GHz; modulation frequency, 100 kHz; field modulation amplitude, 2 G; time constant, 10 ms; scan time, 41s (5 scans). Distilled and deionised water was used for the preparation of DMSO/water solutions. The spin trapping agent 5,5-dimethyl-1-pyrroline *N*-oxide (DMPO; Sigma-Aldrich, Schnelldorf, Germany) was distilled prior to the application.

### 2.11. Western Blotting Experiments

The experiments were performed, as described previously [21].

## 3. Results and Discussion

### 3.1. Synthesis and Characterisation of the **HL^1^**, **HL^2^**, Cu(II) Complexes **1** and **2** and Fe(III) Complex **3**

**HL^1^** was obtained by the condensation reaction of 3-*N*-(*tert*-butyloxycarbonyl)amino-pyridine-2-carboxaldehyde with 2-pyridinamidrazone in the presence of 12 M HCl in 1:1:1 molar ratio in ethanol followed by Boc-deprotection in 4 M HCl/ethyl acetate (EtOAc) (1:6) at 45 °C in 74% yield (Appendix A). The reaction was monitored by positive ion ESI mass spectrometry. First, the formation of **Boc-HL^1^** was confirmed by the presence of the peak at *m*/*z* 341.16 attributed to [Boc-HL^1^+H]^+^, and then the Boc-deprotection by the presence of the peak at *m*/*z* 241.23 assigned to [HL^1^+H]^+^. It was noticed that Boc-deprotection performed under more severe conditions by prolonged boiling the **Boc-HL^1^** in the presence of excess 12 M HCl (1:5) in ethanol/water (3/1) afforded another product (**HL^1a^**) with the same molecular mass as **HL^1^** as confirmed by ESI mass spectrum (see Supporting Information (SI) for the synthesis of **HL^1a^** and Appendix A for its line drawing). The formation of two different products **HL^1^** (crystallisation in ethyl acetate/hexane) and **HL^1a^** (crystallisation from mother liquor as **[H_3_L^1a^]Cl_2_·2H_2_O**) was confirmed by single crystal X-ray diffraction analyses (see Figure 1 and Appendix A) and multinuclear NMR spectroscopy.

It was envisioned that **HL^1^** may exist in two tautomeric forms **A^HL1^** and **B^HL1^** (Appendix A). However, the ^1^H NMR spectrum of **HL^1^** in DMSO-d_6_ showed only one set of signals, which was attributed to the tautomeric form **A^HL1^**. The presence of two amine groups was evidenced by two proton singlets at 6.85 (H_16′_) and 7.10 (H_7′_) ppm in the ^1^H NMR spectrum, as well as by the N-H resonances at 71.27 (N_16′_) and 76.02 (N_7′_) ppm in the ^1^H^15^N HSQC spectrum (for atom numbering see Appendix A).

The reaction of **HL^1^** with CuCl_2_·2H_2_O in methanol in 1:1 molar ratio under reflux afforded a brown solid that was purified by preparative HPLC to give **1·H_2_O** (Appendix A). The positive ion ESI mass spectrum showed peaks at *m*/*z* 338.14 and 303.17 which could be assigned to [Cu^II^(HL^1^)Cl]^+^ and [Cu^I^(HL^1^)]^+^, respectively (Appendix A). The X-ray diffraction quality single crystals of **1** were grown in methanol. Several attempts to prepare the Fe(III) complex with **HL^1^** in a similar manner failed.

**HL^2^** was synthesised by the reaction of 3-N-(*tert*-butyloxycarbonyl)amino-2-pyridinecarboxaldehyde with S-methylisothiosemicarbazidium chloride in the presence of excess hydrochloric acid followed by neutralisation of the reaction mixture with NaHCO_3_ in water in 67% yield. The positive ion ESI mass spectrum showed a peak at *m*/*z* 210.06 attributed to [HL^2^+H]^+^. Single crystals of **HL^2^** were obtained from ethanol/water (3/1) mixture. The 2D NMR spectra of **HL^2^** (in DMSO-d_6_) contain two sets of signals in 1.6:1 intensity ratio, which were attributed to thioamide (form **A^HL2^**) *E*/Z-isomers (Appendix A). The *E*-isomer is stabilised by the intramolecular N–H···N hydrogen bond (Appendix A). In accord with this, the proton resonance of H_7_ is down-field shifted to 6.87 ppm (dominant set, *E*-isomer); H_7_ in the minor isomer is seen at 6.67 ppm (*Z*-isomer). As for **HL^1^**, the presence of two amine groups was confirmed by two proton singlets displayed by *E*-isomer at 6.87 (H_7_) and 6.82 (H_13_) and *Z*-isomer at 6.90 (H_13_) and 6.67 (H_7_) ppm, as well as by the N-H resonances for *E*-isomer at 66.7 (N_7_) and 80.5 (N_13_) or for *Z*-isomer at 70.9 (N_7_) and 83.7 (N_13_) ppm in the ^1^H^15^N HSQC spectra.

The reaction of **HL^2^** with CuCl_2_·2H_2_O in methanol in a 1:1 molar ratio afforded dark-green crystals of **2** of X-ray diffraction quality. The positive ion ESI mass spectrum showed peaks at *m*/*z* 271.15 and 307.13 which could be assigned to [Cu^II^(L^2^)]^+^ and [Cu^II^(HL^2^)Cl]^+^, respectively. Black microcrystalline product **3·0.75H_2_O** was prepared from **HL^2^**, Fe(NO_3_)_3_∙9H_2_O and triethylamine in 2:1:2 molar ratio in methanol. The positive ion ESI mass spectrum showed a peak at *m/z* 472.20, which could be assigned to [Fe(L**^2^**)_2_]^+^ (Appendix A).

### 3.2. X-ray Crystallography

The results of X-ray diffraction studies of **HL^1^**, **[H_3_L^1a^]Cl_2_·2H_2_O**, **HL^2^**, **[Cu(HL^1^)Cl_2_]** (**1**) and **[Cu(HL^2^)Cl_2_]** (**2**) are shown in Figure 1 and Appendix A. The free ligand **HL^1^** crystallised in the orthorhombic space group *Pna*2_1_, **HL^2^** and **1** in the monoclinic space group *P*2_1_/*c*, while **2** in the triclinic centrosymmetric space group *P*$\bar{1}$. 

Both, **HL^1^** and **HL^2^** in **1** and **2** act as neutral tridentate ligands. **HL^1^** forms one six-membered and one five-membered chelate rings upon coordination to Cu(II), while **HL^2^** forms two five-membered chelate rings. The coordination polyhedron around Cu(II) in **1** is intermediate between trigonal-bipyramidal and square-pyramidal (τ_5_ = 0.58). The same distorted coordination geometry was reported previously for a closely related complex [Cu(appc)Cl_2_] (appc = 2-acetylpyridine-pyridine-2-carboxamidrazone) (τ_5_ = 0.58) [45]. The coordination geometry of copper(II) in **2** in contrast is very close to square-pyramidal (τ_5_ = 0.11) [46]. The tridentate ligand **HL^2^** is bound to Cu(II) via nitrogen atoms N1, N3 and N5. The coordination polyhedron is further completed by two chlorido co-ligands Cl1 and Cl2. The configuration adopted by the ligand in **2** differs considerably from that adopted by **HL^2^** in the solid-state. A two-fold rotation of the pyridine ring around C5–C6 bond and of the isothioamide unit around N4–C7 bond in **HL^2^** are required to achieve the configuration of the isothiosemicarbazone ligand in **2**. As in other Cu(II) complexes with tridentate isothiosemicarbazones [34], the thiomethyl group is not involved in coordination to the 3d transition metal atom.

### 3.3. Solution Chemistry of **HL**^1^, **HL**^2^ and Their Cu(II) Complexes (**1** and **2**)

The solution behaviour of Cu(II) complexes with triapine and other pyridine-2 carboxaldehyde thiosemicarbazones has been reported in our recent works [19,21,47,48], revealing the predominant formation of [CuL]^+^ complexes in a wide pH range, including the physiological pH, where L is the deprotonated monoanionic ligand. The stability of this type of complex with the (N_pyridine_,N,S^−^) donor set is so high that its decomposition is negligible even at low micromolar concentrations. However, the solution stability data was never reported for the carboxamidrazone and the pyridine-2-carboxaldehyde S-methylisothiosemicarbazone free ligands and their Cu(II) complexes. **HL^2^** and **2** were selected for a more detailed solution study, since the aim was to reveal the solution stability differences between Cu(II) complexes of the isothiosemicarbazone “S-methyltriapine” **HL^2^** with (N,N,N) donor set and the thiosemicarbazone (triapine) with (N,N,S) donor atoms. In addition, the solution chemical properties of **HL^1^** and **1** were also monitored by UV–Vis spectrophotometry. Firstly, proton dissociation processes of **HL^2^** were followed by pH-potentiometry in water (Appendix A). Notably, **HL^1^** could not be studied by this technique, due to its limited water-solubility. Based on the recorded titration curves of **HL^2^**, p*K*_a_ = 4.96 ± 0.05 was determined, which can be attributed to the deprotonation of the pyridinium nitrogen. This value is higher than that for triapine (4.25) [48]. The hydrolytic stability of this free ligand was checked by a second titration with KOH following back-acidification of the initially titrated sample, and a similar p*K*_a_ (5.05 ± 0.08) was obtained. p*K*_a_ values (p*K*_1_ ~ 2.4 and p*K*_2_ ~ 3.5) for **HL^1^** could be only estimated via spectrophotometric titrations (Appendix A), since the spectra revealed time-dependence. To further investigate the solution stability of both ligands, UV–Vis spectra were recorded under argon at physiological pH using HEPES buffer (Figure 2) and demonstrated significant changes with time indicating the possible decomposition of the compounds. In case of **HL^2^** most likely the elimination of CH_3_SH takes place in agreement with the appearance of the characteristic mercaptan odour of the samples and it seems that this transformation has no marked effect on the p*K*_a_ of the pyridinium nitrogen. The time dependence of the absorbance spectra was also followed at pH 2.2 (HCl) and 9.8 (NaOH), showing similar spectral changes. However, the process becomes faster with increasing pH (Appendix A), while it was significantly slower in pure water at pH 6.3 (without electrolyte and buffer components). The spectral changes of **HL^1^** (Appendix A) were reminiscent of those of **HL^2^**, although the rate of the reaction was the slowest at pH 7.4. In this case, the significant decrease of the absorbance band at 374–400 nm is most likely the consequence of the less extended conjugation system in the molecule, due to the cleavage of one of the C=N Schiff base bonds. This type of hydrolysis is typically catalysed by acid or base. Additionally, the Cu(II) binding ability of both ligands was considerably diminished confirming their partial decomposition as a large difference in the spectra of samples containing free ligands kept at acidic pH (for 0 and 24 h) prior to the addition of Cu(II) ions and adjusting the pH to 7.4 (Appendix A). Although the spectral changes are relatively slow in all cases based on these findings, the complete wide pH range solution equilibrium studies could not be performed.

The solution stability of **1** and **2** was also investigated, and time-dependent UV–Vis spectra recorded at pH 2, 7.4 and 10 (Appendix A) demonstrated only minor spectral changes, indicating that the complexes are more robust at physiological pH than their respective ligands.

Next, the thermodynamic stability of complexes **1** and **2** was investigated by competition reactions with EDTA at pH 5.9 (Figure 3 for **2**). This pH was chosen as comparable conditional stability constants are available for analogous complexes. Representative UV–Vis spectra are shown in the wavelength range where only **2** and the free ligand absorb light. High excess of EDTA was needed to achieve measurable ligand replacement, and log*K*’_5.9_ = 7.38 ± 0.02 as conditional stability constant for **2** was obtained, and in case of **1** log *K*’_5.9_ = 8.20 ± 0.02 was determined. These constants are lower than that of triapine (12.88), ref. [48] implying the weaker Cu(II) binding ability of both **HL^1^** and **HL^2^** compared to the pyridine-2-carboxaldehyde thiosemicarbazones. 

### 3.4. Cytotoxicity

Complexes **1–3** were tested for antiproliferative activity against two cancer cell lines—ovarian carcinoma A2780 and its cisplatin-resistant analogue A2780cis, as well as noncancerous human embryonic kidney cell line HEK293. Their cytotoxicity, reflected by 50% inhibitory concentration (IC_50_) values, was compared to that of triapine, cisplatin and respective free ligands (Table 2 and Appendix A). The differences in the cytotoxicity between cisplatin-sensitive and cisplatin-resistant cell lines are represented by resistance factors (RF). The selectivity to cancer cells over healthy noncancerous cells is represented by the selectivity factor (SF).

From Table 2, it is seen that the free ligands **HL^1^** and **HL^2^** were only marginally cytotoxic, and that their coordination to Cu(II) or Fe(III) resulted in a significant improvement of cytotoxicity, 14- and 144-fold change for Cu(II) complexes **1** and **2**, respectively, and nearly 5-fold increase for Fe(III) complex **3**. Complex **2** was more active than triapine and cisplatin in cisplatin-resistant A2780cis cell line. In general, all compounds were similarly cytotoxic in both A2780 and A2780cis cells, which is reflected by their low RFs. Unfortunately, these compounds also demonstrated comparable toxicity in noncancerous HEK293 cells, similar to triapine and other previously reported thiosemicarbazones [21].

## 4. Investigation of Mechanism of Action

### 4.1. Intracellular Accumulation

It is known that CuCl_2_ salt displays anticancer activity in a range of cancer cells lines, which may be attributed to intracellular Cu overload which leads to ROS production, and altered cancer cell signalling [49,50]. However, despite the disruption of the natural homeostasis of this endogenous metal ion, the cytotoxicity of CuCl_2_ is very limited and manifested only at high micromolar concentrations. This is related to the hydrophilic nature of the inorganic salt, which prevents its efficient uptake by the cells. Since both **HL^1^** and **HL^2^** did not show any reasonable anticancer activity, it was anticipated that the improved cytotoxicity of the metal complexes was directly dependent on the metal centre. In order to compare the intracellular Cu accumulation between **1**, **2** and CuCl_2_, Cu content in A2780 cells treated with these compounds (using 0.3 × IC_50_ concentration) was measured by ICP-MS (Table 2 and Figure 4). Cellular accumulation of the drugs depends on the cell membrane permeability and is known to increase when cellular membranes are compromised. Therefore, cells were treated with the non-toxic equipotent concentrations of the drugs with respect to their killing potential. In agreement with the initial hypothesis, both complexes demonstrated similar Cu accumulation as CuCl_2_ (0.64–0.76 nmol Cu/mg protein). Similarly, for IC_50_ concentrations, complexes **1** and **2** also demonstrated non-significant differences in Cu accumulation (1.1 ± 0.3 and 0.76 ± 0.17 nmol Cu/mg protein for **1** and **2**, respectively). This result indicates that despite significant differences in the cytotoxicity of complexes **1**, **2** and CuCl_2_, the active Cu concentration in drug-treated cancer cells is similar for all three compounds, suggesting that cytotoxicity may be determined by the efficacy of intracellular delivery of Cu ions. However, it should be emphasised that almost identical Cu concentration was achieved when the concentration of **2** was ca. 12, and 60 times lower than the concentrations of **1,** and CuCl_2_, respectively. The improved cytotoxicity of **2** may be related to a more efficient intracellular delivery or a less efficient cellular efflux. 

### 4.2. Electrochemistry

Both, Cu and Fe are redox-active metals, which are essential for the normal cellular function, and their overload triggers the integrative cellular response, including oxidative and proteostatic stress. Since the role of the metal ion in the anticancer activity of complexes **1** and **2** was evident (see Table 2), complexes 1–3 were investigated for their ability to release metal ions within the biologically accessible window, which may further lead to the induction of cellular stress. To investigate the redox properties of 1–3, a detailed electrochemical study was performed by cyclic voltammetry, as well as UV–Vis–spectroelectrochemistry.

The cyclic voltammogram of **1** in DMSO/*n*-Bu_4_NPF_6_ at Pt working electrode showed one irreversible reduction peak with cathodic peak potential *E*_pc_ = −0.76 V vs Fc^+^/Fc^0^ (Figure 5a). The first irreversible reduction step can be attributed to the Cu(II) → Cu(I) process, followed by ligand release from the unstable Cu(I)L complex. Similar to **1**, complex **2** decomposed upon reduction; however, even more intricate redox behaviour was observed with two irreversible cathodic waves at *E*_pc_^1^ = −0.67 V and *E*_pc_^2^ = −1.05 V vs Fc^+^/Fc^0^ (Figure 5b). To further explore the chemical processes occurring at the first reduction step of **2**, the in situ spectroelectrochemical UV–Vis and cyclic voltammetric experiments were carried out under an argon atmosphere in a special thin layer spectroelectrochemical cell with a microstructured honeycomb working electrode. The UV–Vis spectra measured upon cathodic reduction of **2** revealed a new optical band at 373 nm in the region of the first reduction peak, as well as a decrease of the intensity of the two bands at 284 and 448 nm (Figure 6a). Upon scan reversal, the product that was formed upon reduction of Cu(II) to Cu(I) was not reoxidised back to the initial state (Figure 6b). This unambiguously indicates that Cu(I) species were not stable and irreversibly decomposed with the full or partial ligand release. On the contrary, ferric complex **3** exhibited different redox behaviour with almost reversible electrochemical reduction with the first reduction event at *E*_1/2_ = –0.81 V vs Fc^+^/Fc (Figure 5c).

### 4.3. Intracellular Release of Metal Ions

Cancer cells are characterised by the high levels of GSH, the low molecular mass antioxidant tripeptide, and a reducing intracellular environment. Following the observations from the cyclic voltammetry experiments, it was investigated whether complexes **1** and **2** could release Cu ions in the presence of GSH. The reduction of **1** and **2** by GSH was followed by UV–Vis spectrophotometry under the anaerobic condition at pH 7.4. A tightly closed tandem cuvette was used, containing Cu complexes in one of the pockets, and 120 equiv. GSH in the other one. After triggering the reaction, the first recorded spectrum (10 s) revealed significant spectral changes, due to the fairly fast reduction of the complexes (Appendix A), similar to the spectroelectrochemical UV–Vis behaviour summarised in Figure 6a. The reaction was found to be extremely fast for **1** and the final spectrum corresponded to that of the free ligand **HL^1^**. However, the generated final spectrum in case of **2** does not correspond to the ligand **HL^2^**, the λ_max_ (388 nm) is somewhat shifted to the higher wavelengths (Appendix A, λ_max_ of **HL^2^**: 362 nm under the same condition). When passing oxygen through the solutions of the reduced complexes, there were no significant spectral changes observed, and the initial spectra could not be recovered.

Upon cathodic reduction of **3** in DMSO/*n*-Bu_4_NPF_6_ at the first electron transfer new absorption bands at 649, 395 and 289 nm arise with a simultaneous decrease of the initial optical band at 500 nm via isosbestic points at 550, 444, 274 and 300 nm (Figure 7a). Additionally, upon voltammetric reverse scan nearly full recovery of the initial optical bands was observed, attesting the chemical reversibility of the cathodic reduction even at low scan rates (Figure 7b). These results are comparable with those reported for other Fe(III)–TSC complexes [27,51].

Based on the results of the spectroelectrochemical experiments, it was concluded that complexes **1**–**3** can release metal ions in the reducing cellular environment. However, even minor changes in the structure of the complexes significantly affected chemical transformations accompanying their reduction. The reduction of complex **1** resulted in the irreversible release of inactive ligand **HL^1^** and Cu ions; therefore, its anticancer activity may be fully attributable to the Cu centre. Similarly, the reduction of complex **2** also resulted in an irreversible release of Cu ions; however, it was accompanied by the formation of a new organic species, different from **HL^2^**, which has not been identified. On the contrary, Fe complex **3** demonstrated reversible redox cycling, which commonly leads to sustainable cellular ROS production.

*Ex vivo ROS detection.* To further investigate the redox processes occurring upon reversible reduction of complex **3**, we performed EPR spin trapping experiments with [**Fe^II^(L^2^)_2_]**, in comparison with the Fe^II^-triapine complex, **Fe^II^(3-APH_−__1_)_2_**, ref. [10] using DMPO as the spin trapping agent (Appendix A). The reaction mixture contained 0.45 mM free ligand in 1% (*v*/*v*) DMSO/H_2_O, 0.04 M DMPO, freshly prepared aqueous solution of FeSO_4_ (0.22 mM) to form bis-ligand complexes, and 0.1 M H_2_O_2_, and the EPR spectra were recorded immediately. As shown in Appendix A, for both, [**Fe^II^(3-APH_−__1_)_2_]** and [**Fe^II^(L^2^)_2_]** the rapid formation of the DMPO-OH• spin-adducts with similar yields was confirmed in both cases. No production of ROS was observed for free ligands (3-AP and **HL^2^**), confirming the crucial role of Fe for ROS generation. More stable adducts were observed in 5% (*v*/*v*) DMSO/H_2_O system where the carbon centred spin-adducts dominated, as previously reported for dithiothreitol-reduced [**Fe^II^(3-APH_−__1_)_2_]** in the presence of DMPO [11]. In this case, DMSO acted as a HO• scavenger generating methyl radicals which are trapped by DMPO. The EPR spin trapping experiments suggest that reversible reduction of complex **3** led to the production of the Fenton-generated OH• radicals, similar to the Fe^II^-triapine complex, which may be responsible for cancer cell death.

### 4.4. hR2 RNR Inhibition by ***HL**^2^*

The results presented thus far, regarding the lead compound, complex 2, showed that **HL^2^** coordination to Cu(II) resulted in a significant improvement of its cytotoxicity, which is most likely determined by its efficient intracellular delivery of copper ions. To further confirm that the cytotoxicity of **2** is determined by the ligand’s ability to serve as a copper delivery vehicle, further experiments with the ligand, **HL^2^**, were performed. Namely, since **HL^2^** is structurally related to 3-AP, which has been shown to be a potent hR2 RNR inhibitor, tyrosyl radical destruction in hR2 by **HL^2^** was measured by EPR spectroscopy at 30 K and compared to that by 3-AP. 3-AP efficiently inhibits mammalian R2 RNR in the presence of an external reductant (GSH or DTT), and as reported recently [21], several studies have shown that it is the formed [**Fe^II^(3-APH_–1_)_2_]** species that is responsible for the R2-specific RNR inhibitory effect of 3-AP [9,10,11,52,53].

When equimolar amounts of protein and ligand were incubated in the presence of DTT, 50% tyrosyl radical reduction was observed after 45 s with 3-AP, and 5 min with **HL^2^** (Figure 8). Furthermore, 3-AP reduced 100% radical after approximately 3 min, which is in agreement with previously published results for mouse R2 RNR [10]. However, in the same experimental conditions, **HL^2^** was not able to reduce 100% radical in hR2 even after 10 min. These results indicate that **HL^2^** is not as efficient hR2 inhibitor as 3-AP, in agreement with its marginal cytotoxicity, and provides further evidence about its role as an intracellular copper delivery vehicle, contributing to the cytotoxic activity of complex **2**.

### 4.5. The Effects of Cu Overload on Cancer Cell Signalling

While ROS-inducing properties of Cu ions, as well as Cu complexes, have been exhaustively studied, their effects on cancer signalling cascades have been largely overlooked [54]. Recently, the microarray analysis of Cu-overloaded colon cancer cells revealed the disruption of genes involved in unfolded protein response (UPR), proteasomal degradation and autophagy—the major pro-survival cellular responses [49]. However, prolonged Cu treatment led to the activation of caspase-dependent and -independent modes of cell death [49]. Several Cu-thiosemicarbazone complexes were also shown to induce UPR, autophagy and antioxidant defence [21,55,56]. Therefore, the effects of the most active complex **2** on the expression of pro-apoptotic UPR marker CHOP were determined, as well as the antioxidant defence marker Nrf2 (Figure 9). When A2780 cells were treated with increasing concentrations of **2** for 24 h, the increase of both, Nrf2 and CHOP was observed even at low concentrations (1 μM). In addition, the drug treatment interference with cell cycle was investigated by monitoring the markers playing an important role in the cell cycle progression through gap phases, namely, cyclin D1 (G_1_/S) and cyclin B1 (G_2_/M). While no changes in cyclin D1 expression were observed, the decrease in cyclin B1 expression indicated the cell halt at G_2_/M phase and delayed progression to mitosis. This observation was in stark contrast with other Cu-TSC complexes which caused cell cycle arrest at G_1_/S characterised by the decreased expression of cyclin D1 [21]. Finally, the effects of complex **2** on the PARP cleavage were investigated, which is a key event occurring during apoptosis. As expected, the expression of total PARP decreased, while the expression of cleaved PARP increased in a dose-dependent manner, indicating that complex **2** induced apoptotic cell death in A2780 cells, apoptotic cell death in A2780 cells.

## 5. Conclusions

While the entrance of triapine as an anticancer agent into clinical trials has revealed its limitations, due to side effects and ineffectiveness against solid tumours, the class of α-N-heterocyclic thiosemicarbazones, and their metal complexes, have shown promising potential for further development as anticancer drugs. Herein, two triapine derivatives **HL^1^** and **HL^2^** were prepared, which act as neutral N,N,N-tridentate ligands in their Cu(II) complexes, in contrast to triapine, which coordinates first-row transition metals through N,N,S donor atoms. The spectroelectrochemical experiments indicated that Cu complexes **1** and **2** underwent irreversible reduction of Cu(II) centres accompanied by the release of the free ligands and Cu(I) ions. On the other hand, Fe(III) complex **3**, where **HL^2^** acts as monoanionic ligand, was shown to reversibly redox cycle leading to the production of the hydroxyl radical. The lead complex **2** demonstrated comparable cytotoxicity to triapine, but was less toxic to healthy cells. Based on the mechanistic investigations in ovarian cancer cells, it is likely that the non-cytotoxic **HL^1^** and **HL^2^** play an important role in the delivery of Cu, resulting in the disruption of the tightly regulated intracellular balance of this essential metal ion. Thus, **HL^1^** and **HL^2^** act as effective vehicles for intracellular Cu delivery and may be of interest for future therapeutic development.

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
