# Peer review of "Triapine Derivatives Act as Copper Delivery Vehicles to Induce Deadly Metal Overload in Cancer Cells"

_biomolecules, 2020, doi:10.3390/biom10091336_

Round 1
Reviewer 1 Report
The paper presented by Ohui and co-workers describes the synthesis, full analysis and investigation of antiproliferative activity of novel trapine derivatives. In my opinion, the presented work may have a significant scientific impact, manuscript is well written and prepared however, some data should be better presented.
My comments are presented below. Please provide the explanation for all of them, make changes in the text.
Major concerns:
- Abstract – in my opinion abstract should better presents the aim of the work, the short introduction (one or two sentences) is needed.
- Introduction – page 3, line 72 – briefly describe the pointed problem, add some sentences into the text.
- In my opinion UV-Vis notation should be used instead of UV-vis
- Page 4, chart 1 – chart 1 is not mentioned in the text. Reference to the chart 1 should be presented in the text. It may facilitate the text interpretation.
- Experimental section – according to the Instructions for Authors the name of this section should be changed into the Materials and methods.
- check the notation of hydrates and hydrochlorides, correct it into the i.e. • H2O form.
- page 5, line 123 – acetate/hexane mixture (1/1) – it should be acetate/hexane mixture (1/1 v/v)
- page 7, physical measurements – determine the ESI-MS measurement parameters (parameters of the mass spectrometer including m/z range, scan mode, temperatures, heating gas, CID gas, potential, voltage…)
- Authors used cell line so I think that the agreement of the local ethical committee is needed.
- page 12, results and discussion – according to the instruction for authors this section should be divided into the Results and Discussion sections, there is a lot of data.
- page 12, peak with m/z … - according to mass spectrometry notation it will be better to change it into – signal at m/z or peak at m/z.
- see the reference style, check it and correct according to the journal’s instructions.
- there is a lot of NMR, mass spectrometry and other data presented in the text which presents how a great job the Authors did. However, in my opinion, the MS and NMR data (spectra) should be presented in the supplementary data.
Author Response
Reviewer #1:
The paper presented by Ohui and co-workers describes the synthesis, full analysis and investigation of antiproliferative activity of novel triapine derivatives. In my opinion, the presented work may have a significant scientific impact, manuscript is well written and prepared however, some data should be better presented.
My comments are presented below. Please provide the explanation for all of them, make changes in the text.
Major concerns:
- Abstract – in my opinion abstract should better present the aim of the work, the short introduction (one or two sentences) is needed.
Response: As suggested by this reviewer two sentences have been added to better present the aim of this research: “Thiosemicarbazones continue to attract the interest of researchers as potential anticancer drugs. 3-Aminopyridine-2-carboxaldehyde thiosemicarbazone or Triapine, is the most well-known representative of this class of compounds which entered multiple phase I and II clinical trials”.
- Introduction – page 3, line 72 – briefly describe the pointed problem, add some sentences into the text.
Response: A possible rationale underlying the lower antiproliferative activity of Cu(II) and Fe(III) complexes with 3-AP when compared to that of 3-AP itself was added to the text.
- In my opinion UV-Vis notation should be used instead of UV-vis
Response: We usually use the notation UV–vis. Nevertheless, we agreed to use the notation suggested by this reviewer and performed this change throughout the text.
- Page 4, chart 1 – chart 1 is not mentioned in the text. Reference to the chart 1 should be presented in the text. It may facilitate the text interpretation.
Response: Thank you for catching this mistake. We made a reference to chart 1 in the text on page 3 now.
- Experimental section – according to the Instructions for Authors the name of this section should be changed into the Materials and methods.
Response: Thank you for this suggestion. The Experimental Section was re-named as suggested by this reviewer.
- check the notation of hydrates and hydrochlorides, correct it into the i.e. • H2O form.
Response: We do not agree with this suggestion. We would like to keep the notation we used throughout hundreds of papers published.
- page 5, line 123 – acetate/hexane mixture (1/1) – it should be acetate/hexane mixture (1/1 v/v)
Response: amended.
- page 7, physical measurements – determine the ESI-MS measurement parameters (parameters of the mass spectrometer including m/z range, scan mode, temperatures, heating gas, CID gas, potential, voltage…)
Response: ESI-MS measurement parameters have been added now on p. 7 in the section Physical Measurements.
- Authors used cell line so I think that the agreement of the local ethical committee is needed.
Response: We published more than 150 papers, in which data on antiproliferative activity assays in cancer cells were reported. There are no any ethical issues in this respect.
- page 12, results and discussion – according to the instruction for authors this section should be divided into the Results and Discussion sections, there is a lot of data.
Response: If this reviewer does not insist, we would like to keep this section as it is. In our opinion this is reasonable and in agreement with the opinion of two other reviewers.
- page 12, peak with m/z … - according to mass spectrometry notation it will be better to change it into – signal at m/z or peak at m/z.
Response: amended.
- see the reference style, check it and correct according to the journal’s instructions.
Response: The list of references was re-formated by using ZOTERO.
- there is a lot of NMR, mass spectrometry and other data presented in the text which presents how a great job the Authors did. However, in my opinion, the MS and NMR data (spectra) should be presented in the supplementary data.
Response: In our opinion these data are quite important for identification and characterization of the compounds presented in the manuscript, and, therefore, we would like to keep them in the manuscript. Moreover, NMR spectra are reported only for two compounds, while quoting of m/z values does not need much extra space.
Reviewer 2 Report
Manuscript submitted for revision is interesting. A several issues should be explained:
1) In my opinion, it would be better if the control cell line was healthy ovarian epithelial cells rather than the human embronic kidneys cells. Please answer the question: why was ovarian carcinoma cells selected for the experiment and what is there a rational explanation for this decision?
2) The subsection entitled: "hR2 RNR inhibition by HL2" is redundant. The aim of the research was to determine the mechanism of activity of the complexes (marked 1, 2, 3) and not the HL2 ligand, which was used to create the complex. The sentence: "These results indicate that HL2 is not as efficient hR2 inhibitor as 3-AP, in agreement with its marginal cytotoxicity" is correct but do not explains the mechanism of action of the complexes. It would be better if the same experiment was performed for complex 1 or 2.
3) Densitometry of the bands visualized on the gel are the standard for western blot analysis. In the figure 9, this is missing.
Author Response
Manuscript submitted for revision is interesting. A several issues should be explained:
1) In my opinion, it would be better if the control cell line was healthy ovarian epithelial cells rather than the human embronic kidneys cells. Please answer the question: why was ovarian carcinoma cells selected for the experiment and what is there a rational explanation for this decision?
Response: We agree with the reviewer, the healthy ovarian epithelial cells would have been a better control cell line. Unfortunately, we do not have both this cell line and the resources to maintain primary cells which require special media and supplements.
2) The subsection entitled: "hR2 RNR inhibition by HL2" is redundant. The aim of the research was to determine the mechanism of activity of the complexes (marked 1, 2, 3) and not the HL2 ligand, which was used to create the complex. The sentence: "These results indicate that HL2 is not as efficient hR2 inhibitor as 3-AP, in agreement with its marginal cytotoxicity" is correct but do not explains the mechanism of action of the complexes. It would be better if the same experiment was performed for complex 1 or 2.
Response: Principally the reviewer is right. However, originally we were interested to disclose the effect of thiomethylation on the ability of HL2 to quench the tyrosyl radical when compared to that of 3-AP. R2 protein was shown to possess a binding pocket in close proximity of tyrosyl radical, which well accommodates the molecule of Triapine. Since the ability of HL2 to inhibit R2 protein was markedly lower than that of Triapine we did not perform further experiments with complex 1 or 2. Moreover we do not have R2 protein and this experiment cannot be carried out at the moment. The text on pp. 25-26 has been modified taking into account the issue raised by this reviewer.
3) Densitometry of the bands visualized on the gel are the standard for western blot analysis. In the figure 9, this is missing.
Response: Densitometry of the bands was added to the Figure 9.
Reviewer 3 Report
The present manuscript reports a very interesting work, showing some promising results. I would suggest its publication as it is, considering only very few details as:
- GSH appears without explanation on pag 9
- “The reaction of HL1 with CuCl2·2H2O in methanol in 1:1 molar ratio afforded a brown raw” this is not in agreement with the experimental details. Please check where the error is.
Author Response
Reviewer #3:
The present manuscript reports a very interesting work, showing some promising results. I would suggest its publication as it is, considering only very few details as:
- GSH appears without explanation on pag 9
Response: We have specified now the abbreviation used (GSH) on page 11 and on page 23.
- “The reaction of HL1 with CuCl22H2O in methanol in 1:1 molar ratio afforded a brown raw” this is not in agreement with the experimental details. Please check where the error is.
Response: We replaced the word “raw” by “solid” to be in agreement with experimental details (see page 13).
Round 2
Reviewer 1 Report
In the revised version of the presented manuscript Authors presents responses for all of my questions and comments. Manuscript may be accepted for publication after moderate English changes.
Reviewer 2 Report
In my opinion, the major concerns that I pointed out in my first review have been satisfactorily addressed in this revised version. Therefore, I recommend the manuscript for publication in Biomolecules.